# Adapting and Implementing a Blended Collaborative Care Intervention for Older Adults with Multimorbidity: Quantitative and Qualitative Results from the ESCAPE Pilot Study

**DOI:** 10.3390/bs15010079

**Published:** 2025-01-17

**Authors:** Josefine Schulze, Dagmar Lühmann, Jonas Nagel, Cornelia Regner, Christine Zelenak, Kristina Bersch, Christoph Herrmann-Lingen, Matthew M. Burg, Birgit Herbeck-Belnap

**Affiliations:** 1Department of General Practice and Primary Care, University Medical Center Hamburg-Eppendorf, 20246 Hamburg, Germany; d.luehmann@uke.de; 2Department of Psychosomatic Medicine and Psychotherapy, University Medical Center Göttingen, 37073 Göttingen, Germany; jonas.nagel@med.uni-goettingen.de (J.N.); cornelia.regner1@med.uni-goettingen.de (C.R.); christine.zelenak@med.uni-goettingen.de (C.Z.); cherrma@gwdg.de (C.H.-L.); birgit.herbeck@med.uni-goettingen.de (B.H.-B.); 3Clinical Trials Unit, University Medical Center Göttingen, 37075 Göttingen, Germany; kristina.bersch@med.uni-goettingen.de; 4German Center for Cardiovascular Research (DZHK), Partner Site Lower Saxony, 37075 Göttingen, Germany; 5Section of Cardiovascular Medicine, Yale School of Medicine, New Haven, CT 06520, USA; matthew.burg@yale.edu; 6Division of General Internal Medicine, Center for Behavioral Health and Technology, University of Pittsburgh School of Medicine, Pittsburgh, PA 15261, USA

**Keywords:** heart failure, multimorbidity, psychological distress, integrated care, care management, patient-centred care

## Abstract

Multimorbidity poses significant challenges for patients and healthcare systems, often exacerbated by fragmented care and insufficient collaboration across providers. Blended Collaborative Care (BCC) is a promising strategy to address care complexity by partnering care managers (CMs) with primary care providers (PCPs) and specialists. This study aimed to adapt and pilot a BCC intervention for patients aged 65+ with heart failure and physical–mental multimorbidity. Our objectives were to assess the feasibility of the study procedures, patient recruitment, participant satisfaction and acceptability, and to identify necessary adjustments for improving intervention delivery. We evaluated goal attainment and intervention fidelity through standardised electronic documentation by CMs, and patient acceptance and satisfaction through semi-structured interviews. A monocentric, one-arm pilot study involved nine patients with a mean of 6.7 contacts with their CM over three months. Patients’ health goals primarily focused on lifestyle changes and psychosocial support. The intervention was generally well-accepted, with no reported negative consequences. Difficulties in establishing working alliances with PCPs were a barrier to effective implementation. The analysis indicated the need for minor procedural adjustments. Next steps include launching the ESCAPE trial, a large randomised-controlled trial across different European healthcare systems and developing strategies to facilitate PCP involvement.

## 1. Introduction

The rising prevalence of multimorbidity, the co-existence of multiple health conditions, presents a major challenge to patients, their families, and healthcare systems. Studies have shown that multimorbidity is especially prevalent in older adults, with more than two-thirds of this population living with two or more chronic conditions ([50]). The impact of multimorbidity on quality of life and health outcomes can be substantial, often resulting in increased functional limitations, chronic pain, and psychological distress ([15]; [30]; [32]; [52]; [58]). People with multimorbidity are at a higher risk of hospital admissions, prolonged hospital stays, and premature death ([7]; [41]). Fragmented healthcare services and a lack of collaboration across providers add to the complexity of managing multimorbidity, leaving patients and their families with the burden of navigating complex care pathways and coordinating care. This challenge is even greater when both physical and mental health conditions are present ([21]; [23]; [35]), potentially affecting patient safety ([42]). Improving care for patients with multimorbidity requires a paradigm shift from traditional siloed, i.e., single-condition-focused, care to integrated care models ([47]). These models aim to overcome the challenges of fragmented care for complex patients by providing coordination across multiple settings and providers ([27]).

The Chronic Care Model (CCM), developed by [57] ([57]), provides a foundational framework for effective chronic disease management through proactive care, self-management support, delivery system design, clinical information systems, and community resources. It promotes a collaborative approach to meet the needs of individuals with chronic conditions. The Expanded Chronic Care Model (ECCM) builds upon the CCM by integrating elements of population health promotion ([3]). The ECCM emphasises preventive strategies, considers social determinants of health outcomes, and encourages community participation as integral components in managing chronic diseases.

Blended Collaborative Care (BCC), drawing on above models, is a promising strategy for delivering care coordination and support for patients with multimorbidity in a cost-effective manner. BCC models employ non-physician care managers (CMs) with a professional healthcare background who work in partnership with both patients’ primary care providers (PCPs) and specialists from other medical fields to facilitate a coordinated, multiprofessional effort to deliver patient-centred care. This approach adopts a holistic view that addresses both mental and physical health conditions (‘blended’), recognising their interconnectedness and the potential for mutual reinforcement ([19]). In addition to managing medical conditions, BCC also addresses social factors influencing patients’ well-being by facilitating referrals to various community resources that promote both engagement and support. It empowers patients by providing tools and resources and encourages active participation in their healthcare. Furthermore, BCC incorporates preventive care strategies and utilises clinical information systems for ongoing monitoring of health outcomes, facilitating regular assessments and proactive interventions.

Randomised-controlled trials testing BCC models in the US have demonstrated improvements in depressive symptoms, diabetes control, and cardiovascular outcomes ([10]; [11]; [24]; [48]). As the first application in the German healthcare system, a randomised feasibility study of BCC for cardiac patients showed reduced distress and risk factors, with 83% of patients expressing high satisfaction with the intervention ([5]).

The ESCAPE Trial (‘Evaluation of a patient-centred biopsychosocial blended collaborative care pathway for the treatment of multimorbid elderly patients’) aims to adapt and test a BCC intervention in five European healthcare systems (Denmark, Germany, Hungary, Italy, and Lithuania). We focused the intervention on the population of older adults (≥65 years) with heart failure and physical–mental multimorbidity, given the poor health outcomes associated with mental comorbidity and the high need for care coordination in this population ([9]; [54]).

In the following, we outline a logic model illustrating the underlying mechanisms of impact of the ESCAPE BCC intervention and present the strategy tailored to the targeted population. In the presented pilot study, we assess the feasibility of study procedures and recruitment, identified necessary adaptations to improve the implementation of the intervention in the target settings, and evaluate participant satisfaction and acceptability of the intervention.

## 2. Materials and Methods

### 2.1. Adaptation of BCC Intervention Models

Based on US and German trial protocols targeting patients with cardiac and mental health conditions ([19]; [20]; [24]), we drafted the ESCAPE BCC intervention. We extracted core BCC intervention elements and presented them in lay language to people with multimorbidity and their informal carers during semi-structured interviews in three of the five study countries. The interviews revealed a wide range of patient preferences regarding the need for health information, frequency of contacts with CMs, and the level of support and coordination needed ([14]). This resulted in an adapted ESCAPE BCC intervention emphasising a patient-centred approach to care with a collaborative care team including the patients, a CM (typically a nurse or health psychologist), the patients’ PCP, and a supervisory specialist team. The aim of this care team is to support the management of the patient’s health, with the option to involve informal carers, community resources, and specialist care as needed. To integrate the ESCAPE BCC intervention with current primary care treatment, we adapted the meta-algorithm for multimorbidity (MAM) from the German Guideline on Multimorbidity ([36]). The MAM aids treatment prioritisation in primary care while considering the risk of harmful trajectories. It facilitates coordination between PCP and CM and establishes a monitoring system by defining patient-specific symptoms and ‘red flags’, such as rapid weight gain or increased fatigue. To accommodate the range of individual preferences, we decided to train CMs in the core BCC components (outlined in Table 1), basic patient contact structure, and communication tools while allowing flexibility to individualise care management contacts.

### 2.2. Logic Model of the Intervention

We developed a process-oriented logic model (see Figure 1) illustrating the ESCAPE BCC intervention components and their hypothesised impact on patient outcomes ([25]; [46]). The logic model integrates contextual factors influencing patients’ quality of life, e.g., multiple health conditions and their treatments, fragmented care, and limited patient resources, such as reduced self-efficacy, which increase their perceived burden (see patient column, Figure 1). The ESCAPE intervention coordinates the PCP treatment plan (formalised in the MAM), patient preferences, and recommendations of the specialist team into a BCC treatment plan aimed at improving quality of life and promoting healthy lifestyles while enhancing healthcare coordination and supporting evidence-based treatment. The model incorporates CM tasks to collaboratively set and pursue manageable patient goals through individually tailored interventions. Furthermore, symptoms and ‘red flags’ are monitored pro-actively (see BCC column, Figure 1).

The intervention is facilitated by CM skills, a comprehensive week-long training in BCC core elements and psychological techniques, a trusting patient-CM relationship, proactive regular patient contacts, collaboration with chronic care providers, regular case review meetings with the specialist team, and an electronic registry guiding CMs in their patient contacts and documenting necessary information for team sharing (see facilitator column, Figure 1).

### 2.3. Design and Setting of the Pilot Test

#### 2.3.1. Procedures

The procedures of the main trial are detailed in the clinical trial protocol ([59]). To assess the feasibility and acceptability of study procedures, including recruitment, data assessment, and intervention components, while maintaining time and cost efficiency during the pilot study, we adapted the clinical trial protocol for the pilot test by: (a) enrolling patients at only one study site in Germany; (b) eliminating the two-month rescreening period and directly including patients without delay; (c) employing a one-armed trial design with only an intervention group; (d) shortening the treatment phase from nine to three months; and (e) using a pilot Excel-based registry while the final web-based care management registry was under development ([6]). Our pilot study was approved by the Ethics Committee of the University Medical Centre Göttingen (vote no. 11/9/21) and adheres to the ethical standards of the Declaration of Helsinki. The study was registered at German Clinical Trials Register (ID: DRKS00027320).

#### 2.3.2. Sample

We carried out the pilot test (target: N = 10) in Germany, where the majority of the recruitment sites for the main ESCAPE trial are located (40% recruitment target). At the Departments for Cardiology and Psychosomatic Medicine at the University Medical Centre Göttingen, we approached hospitalised patients aged 65 and older diagnosed with heart failure for participation in the study. Interested patients who gave their written informed consent were screened for inclusion/exclusion criteria. Additional inclusion criteria were (a) two or more chronic physical comorbidities and (b) psychological distress indicated either by a score >12 on the Hospital Anxiety and Depression Scale (HADS), a self-report measure comprising two seven-item subscales assessing anxiety and depressive symptoms in the past week ([40]; [60]), and/or a diagnosis of a mental disorder. Exclusion criteria were (a) life expectancy of less than one year, (b) communication barriers such as severe hearing impairment, (c) severe mental disorder such as schizophrenia requiring specific psychiatric treatment, (d) being permanently bedridden, and (e) residing in a nursing home.

#### 2.3.3. Data Assessment and Analysis

Baseline assessments included self-reported sociodemographic and clinical characteristics, cross-referenced with medical records. Feasibility of data collection, documentation in an electronic database, response burden, and questionnaire comprehensibility were scrutinised according to the planned protocols for the main trial ([59]). Following the baseline assessment battery, the CM contacted patients by telephone to introduce them to the intervention.

After conclusion of the intervention period, a follow-up assessment battery was administered, including the Working Alliance Inventory-Short Revised version (WAI-SR). It evaluates satisfaction with the patient-CM working relationship and consists of 12 items rated on a 5-point Likert scale and distributed across 3 domains. Total scores were computed as the mean of the domain scores, each ranging from 5 to 20 points ([37]; [39]).

Additionally, we conducted telephone interviews with patients using a semi-structured guide, focusing on satisfaction with the intervention and their perception of its benefits and potential harms. We developed the interview guide by reviewing the core components of the ESCAPE BCC intervention and generating opening and follow-up questions ([18]; [33]). The interview guide (see Appendix A) was designed to be flexible and could be adapted during the interview to allow for more in-depth exploration of patients’ experiences ([26]). Interviews were recorded and transcribed verbatim. Using deductive categories derived from the core elements of the BCC (see Appendix B), the transcripts were then coded following qualitative content analysis ([31]). Analysis was carried out independently by two researchers who met regularly for discussion and agreement on their coding. Moreover, we conducted debriefing sessions with the CMs and the specialist team exploring topics related to the intervention feasibility and acceptability via field notes.

### 2.4. Pilot Test of the Intervention

During the initial 30-to-45 min contact, the CM explained the details of the intervention to the patient and discussed their healthcare preferences, challenges in managing their conditions, bothersome symptoms, and other burdens affecting their physical and mental health. The CM documented this information directly in an Excel registry, which included relevant domains such as sociodemographic data, medical history, current diagnoses and medications, medical device usage, health behaviour, and treatment preferences (refer to Table 2). Next, the CM extracted the MAM report from the registry, which included patient reports of clinical characteristics and symptoms, medication adherence, and treatment preferences, and sent it to the patient’s PCP. In turn, the PCP reviewed and expanded upon this information and shared a comprehensive care plan. The CM and patient then discussed the care plan and agreed on concrete goals. Progress on these goals was documented during each subsequent CM–patient contact. Going forwards, the CM also monitored bothersome symptoms, possible ‘red flags’, and any changes to patients’ medication, health behaviour, and mental health burden. All CM–patient contacts were telephone-based and scheduled bi-weekly, typically lasting 20–30 min.

During case review meeting, the CM presented their patients to a specialist team consisting of a general practitioner, a cardiologist, a psychosomatic specialist, a clinical pharmacologist, and a geriatrician. The team could consult with other specialists as needed (e.g., nephrologist). After reviewing each patient’s care plan, treatment preferences, progress with their goals, and current perceived physical and mental burden, the specialist team could make recommendations regarding evidence-based treatment, additional support in reaching the goals, and assisting CMs in overcoming challenges. If recommendations pertained to the treatment plan (e.g., medication adjustment, laboratory tests), the CM shared this with the patient’s PCP, who retained primary responsibility for the patient’s medical treatment, with the freedom to accept, modify, or reject the recommendations. The specialist team also monitored intervention fidelity through electronic documentation.

## 3. Results

### 3.1. Study Sample

Between October 2021 and June 2022, we approached 44 patients at the departments for cardiology and psychosomatic medicine at the University Medical Centre Göttingen, of whom 10 (23%) expressed interest in participating and consented to screening for eligibility. Nine of these patients met all eligibility criteria and were enrolled in the intervention (Figure 2).

One patient could not be reached after initial contact, and another passed away before the intervention concluded. As a result, we conducted the planned 3-month intervention with seven patients, and the semi-structured interviews with six patients. The sample description and individual health goals are shown in Table 3. At baseline, patients (N = 9) had a mean age of 70.7 years, and five were female. They presented with a range of 7 to 14 chronic conditions (mean: 10.7) and reported moderate psychological distress (mean total HADS score: 16.1).

### 3.2. Delivery of the Intervention and Quantitative Data Analysis

Contact with the patients’ PCPs (or other preferred main provider) was established in all but one case. CMs had between 2 and 20 contacts (median: 6) with their patients. We conducted a total of 12 case review meetings with the specialist team. According to the data from the electronic registry, patients and CMs agreed on two to four health goals (median: three) and, in a shared decision-making discussion, they prioritised the goals, typically focusing on one at a time. Of the 25 established health goals, 14 focused on lifestyle changes (e.g., physical activity, diet), 9 on psychosocial support, and 2 on managing somatic symptoms. All patients who participated for the entire duration could achieve at least one of their goals. At the end of the intervention, responding patients reported a high level of satisfaction with their working relationship with the CM (WAI-SR mean 15.1; range: 11–19.7), consistent with previous research demonstrating elevating WAI-SR scores among patients receiving care management ([8]). Mean psychological distress was slightly decreased at the end of the intervention (from mean total HADS score 16.1 to 15.1), although not all patients could be included at follow-up. Due to the small sample size, we only report descriptive statistics but refrain from conducting inferential statistics.

### 3.3. Patient Perception of the Intervention

#### 3.3.1. Motivation and Reasons for Participation

The analysis revealed various reasons for participating in the study. Patients were generally open to healthcare research, with some having positive past experiences. A common motive was the desire to improve their situation and contribute to research. Challenges in regular care, such as unclear medication instructions, insufficient information on side effects, and the heavy workload of PCPs, which limited their time for coordination efforts, further motivated them to enrol in the study.

#### 3.3.2. Communication with the Care Team

Patients were overall satisfied with the support they received from the ESCAPE care team. They perceived the calls with the CM to be helpful and encouraging and appreciated the interest in their well-being and the proactive contact. Patients also valued the CMs’ flexibility and responsiveness to their needs. They highlighted that the discussion of emotional issues alongside medical issues was also an important factor in establishing a trusting relationship. While many patients expressed a preference for face-to-face meetings, telephone contact was also considered a viable format and, in most cases, more convenient to integrate into daily routines. Due to the brief duration of the intervention, patients suggested to incorporate a follow-up contact to monitor progress and address arising challenges later on. Furthermore, some patients perceived the standardised questions, e.g., regular mental health screenings, as burdensome and wished for more detailed reports regarding the discussions of the CM with the specialist team during case review meetings.

#### 3.3.3. Provision of Evidence-Based Health Information

Patients appreciated the opportunity to obtain information about their conditions and a treatment plan from the CM, including the opportunity to clarify more specific questions on their health with the specialist team (via CM):

‘I always asked about the topics important to me and then I got answers from the experts. I thought it was great because normally, you don’t get access to these people.’

However, some patients expressed a preference for direct contact with a physician, especially at the beginning or end of the intervention when their treatment plan was discussed, or when seeking guidance beyond what the non-physician CM was able to provide.

#### 3.3.4. Care Coordination and Optimisation of Treatment Plans

Although the extent of collaboration and coordination between CMs and other healthcare providers varied, patients rated this aspect highly relevant to their goal achievement. In all cases, CMs actively sought to collaborate with patients’ PCPs and treating specialists to share information and coordinate treatment plans. One patient perceived the collaboration between her CM and PCP as productive and noticed an increased diagnostic effort by her PCP. However, she was concerned that the collaboration may result in an additional burden on her PCP. Another patient shared his experience of managing multiple medications and the advice of the specialist team:

‘It helped me a lot that they talked to the pharmacist about this drug interaction or something like that. I always felt so sick after two hours when I took the pills in the morning that around lunchtime, I had to lie down (…) And now I have the impression that I tolerate it much better.’

Although patients appreciated the recommendations to optimise their treatment plan, some noted that their PCPs’ lack of engagement hindered implementation. They made suggestions to improve collaboration, such as involving other specialist physicians when PCPs were not available and incentivising providers to increase interprofessional communication. While all patients received suggestions from their CMs to access additional resources like dietary or mental health counselling, only a portion followed through, citing either a lack of motivation or perceived appropriateness.

#### 3.3.5. Support in Implementing the Treatment Plan

The patient interviews confirmed that CMs used critical intervention elements, such as goal setting, behavioural activation, and motivational interviewing, to engage their patients in managing their health. In one example, a patient reported how the process of goal setting helped her to gradually resume her activities:

‘She actually helped me a lot in that respect (…) for example with the bicycle. (…) And she said, take it slow at first and just try to go around the house five times and that’s enough with the bicycle for now. I have to say, she kept encouraging me and telling me to take it slow and try it out. I was actually quite grateful.’

#### 3.3.6. Perceived Effects of the Intervention

Patients reported predominantly positive effects of participation in the study. They noted improved physical performance, psychological well-being, and positive changes to their daily lives. Engagement in the study fostered self-reflection and motivation among patients, with no reports of adverse effects. However, several limitations were identified: Some participants struggled with ambitious goals that proved unattainable within the shortened three-month intervention period, leading to frustration. For example, one participant expressed disappointment at not achieving weight loss despite lifestyle changes. Overall, many emphasised the limitation due to the short duration of the intervention and expressed a desire for a longer timeframe.

### 3.4. Perspective of CMs and Specialist Team on the Intervention

CMs observed that their patients were generally cooperative and open to their suggestions. They found their work most rewarding when patients achieved the goals they had agreed upon together. They suggested use of a toolbox to share materials among CMs, e.g., on motivational interviewing techniques or specific health information. Difficulties in reaching PCPs and other healthcare providers led to delays in processing the MAM and establishing a collaborative treatment plan. Providers were often unavailable and had limited time to engage in the study, which made it difficult to coordinate care and led to interventions not being implemented in a timely manner. Although CMs perceived the case review discussions as helpful, members of the specialist team noted that, as most of them were not used to this type of interprofessional collaboration, it took some time before case reviews could be carried out in a time-efficient manner.

### 3.5. Adaptations to the ESCAPE BCC Intervention

Based on the results of the pilot study, we made the following adaptations to the ESCAPE BCC intervention protocol for the main trial:To address the diverse healthcare needs of patients, the main trial introduces a flexible scheme for CMs to support goal setting and monitoring of symptoms and red flags as mandatory elements, while other aspects (e.g., medication, general health behaviour) remain optional. CM training now includes specific guidance on setting SMART goals, with close monitoring by trainers and specialist teams. Intervention fidelity will be ensured through a centralised ’train the trainer’ workshop and by reviewing the documentation in the registry.To enhance transparency between the specialist team, patients, and PCPs, recommendations will be communicated directly to both patients and PCPs. Two reports summarising progress and recommendations will be provided: one mid-intervention (after 4–5 months) and a final report at the end of the intervention, which will include recommendations for continuing the BCC treatment plan. In view of scalability, we refrain from establishing a direct communication between members of the specialist team and patients but encourage the discussion of their recommendations with the patients’ PCP.CMs are encouraged to expand the comprehensive study intervention manual by assembling health information and community resources in a toolbox, which is shared among study sites. Regular meetings among trainers are scheduled to discuss local implementation issues and major challenges faced by CMs (e.g., communication with PCPs, significant mental or somatic health burdens). The results of these discussions will be documented and included in the final version of the intervention manual.We implement a chairperson within the specialist team to facilitate discussions and share registry documentation. Specific guidelines concerning team composition, meeting structure, presentation formats, and recommendation scopes are provided to maintain protocol fidelity.

## 4. Discussion

This study aimed to adapt and pilot existing Blended Collaborative Care (BCC) strategies in an intervention for patients aged 65 years and older with heart failure and physical–mental multimorbidity. The primary objectives were to test the planned patient recruitment and study procedures, assess the feasibility of the adapted BCC intervention, and evaluate patient acceptability and satisfaction through semi-structured interviews. We also monitored the agreement on and attainment of health goals using standardised electronic documentation.

### 4.1. Main Findings

Overall, the pilot study demonstrated the feasibility of the study procedures. Additionally, the study revealed that this patient population shows a high degree of disease complexity. While recruitment was successful, we experienced dropouts due to death and loss of contact, which need to be taken into account in the planning of the main trial. Despite a significantly shortened implementation and training period, we were able to conduct the intervention as developed for the main trial. Patients engaged readily in the intervention, and there were no recorded negative outcomes. Indeed, patients appreciated having regular contacts with their CMs to talk about their health and care concerns, receiving encouraging and patient-centred support, and improved communication with their healthcare providers. Patients were generally open to communicating by telephone, and most found it acceptable to have a non-physician CM as their contact person. However, some identified the lack of collaborative engagement by their PCPs as significant barrier to implementing recommendations for treatment optimisation and expressed concern that the intervention could pose a potential for additional burden to PCPs. The qualitative interviews at the end of the intervention underscored the importance of setting realistic and achievable goals. Moreover, they highlighted the need for a flexible and individualised care management approach to address the diverse needs of patients.

### 4.2. Strengths and Limitations

The conduct of a pilot study is a major strength of the ESCAPE project, as pilot studies are often time consuming and underfinanced and, for this reason, typically neglected or underreported. Nevertheless, they can play a pivotal role in developing and refining complex interventions and study setups before proceeding to large-scale clinical trials, thus, contributing to their overall quality ([22]; [43]). Furthermore, the use of a logic model provides a valuable framework to promote a shared understanding of the individual components of the intervention and their expected impact on patient outcomes ([34]). Due to the limited sample size, one-armed design, and shortened duration of our intervention, we refrained from hypothesis testing in this pilot study ([1]).

As the pilot study population was limited to Germany, there are critical concerns regarding the generalisability of our findings for the main trial, especially given the variations in healthcare systems across the five participating European countries. The diversity of these systems—encompassing differences in care delivery models and levels of integration—adds complexity to the unified implementation of a BCC approach. For instance, while in Germany, direct access to specialists and patients’ freedom to choose any physician ([28]) may hinder CMs from effectively coordinating patient care, in Denmark, patients are required to register with a designated GP practice ([44]), which could foster a more structured relationship that promotes continuity of care. Similarly, Italy’s healthcare system emphasises the role of GPs ([53]), potentially providing a solid foundation for CMs to operate effectively for both countries. In contrast, in Hungary, where the GP gatekeeping function is weak, patients often experience long waiting times for specialist services and poor communication between GPs and specialists ([49]), a situation that underscores the need for a CM figure to improve care coordination. Additionally, the more integrated and collaborative approaches to primary care in Denmark and Lithuania ([29]; [56]) could further support CMs in coordinating care between different healthcare providers.

### 4.3. Practical Implications and Future Directions

To address the disparities across different healthcare models, the main trial will adopt a two-tiered training approach based on a train-the-trainer model. A centralised training programme will facilitate a standardised approach by preparing a team of trainers from different countries while allowing the diversity of healthcare systems to be recognised. These local trainers will then disseminate knowledge within their healthcare systems to CMs, ensuring that country-specific nuances are incorporated into the training process. In addition, centralised continuous fidelity monitoring of the electronic documentation will be implemented to ensure adherence to the ESCAPE intervention protocol and to provide feedback to local trainers to refine practises as needed. This strategy aims to enhance the successful implementation and scalability of the intervention across different healthcare models in the main trial.

Collaborative care, despite promising evidence, has not yet seen widespread adoption in routine care ([2]). However, previous research suggests that the effectiveness of collaborative care increases as it becomes more integrated ([45]; [51]). A crucial factor in the successful implementation of collaborative care models is the role of CMs ([38]). This sentiment is echoed in our findings, as the integration of CMs into patients’ healthcare networks emerged as a key challenge. A significant barrier to effective implementation is the lack of clarity about the role of CMs and their integration within healthcare systems, especially given PCP’s concern about potential role overlaps ([4]; [17]). By bridging the gap between clinical interventions and lifestyle adjustments, CMs guide patients in accessing relevant points within the healthcare system and act as catalysts for their empowerment. This approach fosters a culture of patient-centeredness, prevention, and holistic health management, thereby overcoming the disease-centred approach to care ([12]; [55]). While CMs have successfully contributed to delivering and coordinating care in previous Collaborative Care Models ([16]), they report that supporting patients with psychological distress can be significantly more challenging than their conventional nursing roles ([55]). Therefore, to effectively support patients with physical–mental multimorbidity, it is critical to establish comprehensive educational programmes that equip CMs with skills in psychosocial support, behavioural health strategies, and navigating the complex healthcare landscape. When introducing new collaborative care interventions, the role of CMs should be clearly defined and adequately prepared, particularly regarding their integration into primary care, and they should be supported during implementation to facilitate meaningful collaboration with PCPs ([38]). In the main trial, some sites will test a closer initial cooperation with the PCP offices, which may reduce this issue. Furthermore, the planned health economics analysis in the main trial ([13]) will also examine cost-effectiveness and possible reimbursement procedures. In view of the small sample size and the focus on Germany in the pilot test, we also intend to conduct a comprehensive 360-degree process evaluation of the main trial, utilising electronic documentation, patient-reported experience measures and qualitative interview data from all members of the BCC team, including CMs and PCPs, to obtain results that can be generalised beyond the specific context. Furthermore, the main trial will extend the follow-up period to six months and longer after the end of the intervention to assess meaningful and sustained changes in patients’ health outcomes.

## 5. Conclusions

Our findings contribute to the growing body of evidence on the procedural and structural challenges of adopting BCC in the chronic care of older patients with multimorbidity. The ESCAPE trial provides a unique opportunity to examine the implementation of the BCC model in different European healthcare settings. The results of the pilot study demonstrate the feasibility and acceptability of procedures related to recruitment, screening, data collection, and the adapted telephone-based BCC intervention for older adults with heart failure and physical–mental multimorbidity. Patients valued the regular contact to the CM in the intervention, patient-centred support, and improved communication with healthcare providers facilitated by the intervention. Moving forward with the intervention, adjustments will be made to address challenges such as PCP engagement by implementing clear communication channels and fostering strong partnerships with healthcare providers.

## Figures and Tables

**Figure 1 behavsci-15-00079-f001:**
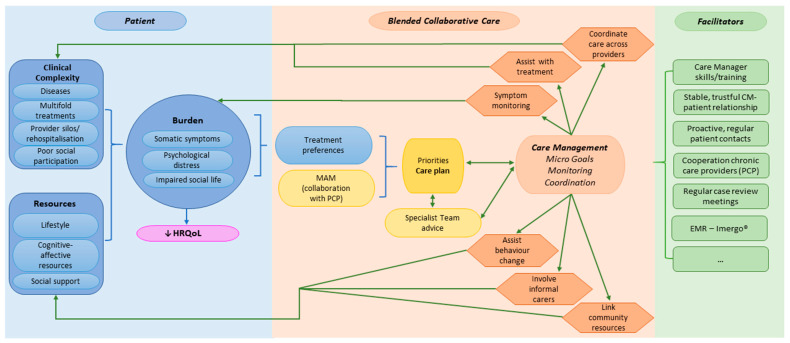
Logic model of the ESCAPE BCC intervention.

**Figure 2 behavsci-15-00079-f002:**
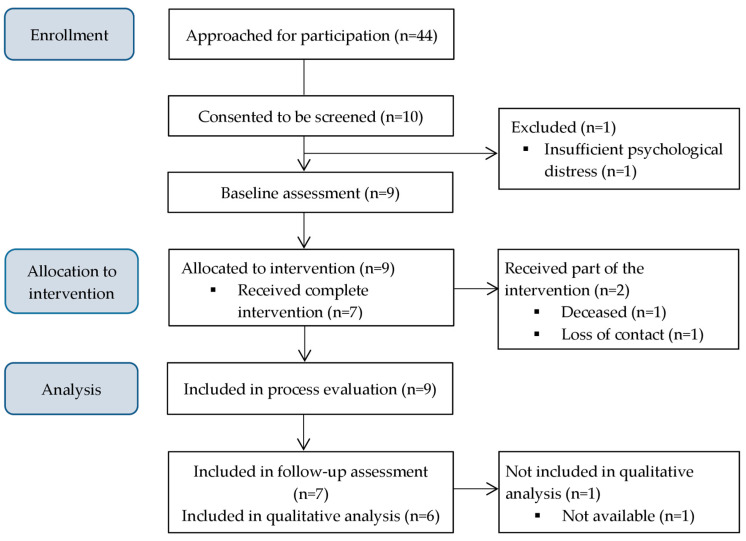
Flowchart of participants’ progress through the phases of the pilot study.

**Table 1 behavsci-15-00079-t001:** Intervention components and care manager tasks.

Intervention Component	Care Manager Tasks
**Individual tailoring of treatment plans**	Obtain medical history through patient assessment, communication with primary care provider (via MAM ^a^) and review of medical letters and electronic clinical records Explore patient priorities and preferences, such as the following: ○ Personal goals, e.g., quality of life, symptom relief, social life, ability to work; ○ Willingness/ability to contribute to health improvement, e.g., exercise, change diet, engage in therapies; ○ Acceptable level of treatment burden; ○ Involvement of informal carers. Coordinate with primary care provider (via MAM) ○ Obtain long-term treatment plan ○ Identify symptoms to be monitored ○ Identify areas where care manager support is needed Agree on health goals using shared-decision making
**Support in translating treatment plan into daily routine**	Regular proactive contact Goal setting using the SMART ^b^ formula and tracking of goal achievement Behavioural activation Address barriers using motivational interviewing and problem-solving techniques Provide self-management options and materials, e.g., dietary log, relaxation techniques Monitor adherence to treatment recommendations as needed
**Provide health information**	Educate about the different conditions and/or provide educational materials ○ General lifestyle and health behaviour education ○ Education on specific conditions (heart failure, psychosocial distress, multimorbidity), e.g., emergency management for heart failure or psychological crisis ○ Specific education on common conditions and their interaction with heart failure/psychological distress, e.g., type II diabetes, hypertension Educate about planned treatments or treatment options, tailored to specific needs
**Monitoring of symptoms**	Monitor for critical symptoms (red flags), e.g., weight gain, syncope Monitor other relevant symptoms and parameters, e.g., blood glucose, pain Monitor emotional distress, e.g., stress burden, levels of depression and anxiety
**Care coordination**	Coordinate with primary care provider Collaborate with carer, if desired Facilitate communication across all treating healthcare providers Assist with access to community resources (self-help groups, volunteer programme) Facilitate specialist referrals Provide resources to address healthcare-related social issues, e.g., transportation, payment for medication Monitor medication prescriptions under specialist supervision for evidence-based efficacy and adverse effects

**NOTES:** ^a^ MAM = meta-algorithm for multimorbidity; ^b^ SMART goals = specific, measurable, attainable, relevant and time-bound goals.

**Table 2 behavsci-15-00079-t002:** Components of the care management registry.

Component	Variables
**Header**	Patient core data, overview of variables to be monitored, list of diagnoses, patient preferences
**Contact overview**	List of contacts with date, presence of red flags, goal attainment, clinical and mental health status
**Care Management ***	List of goals with description of care management interventions and goal attainment, symptom monitoring, red flags
**Patient Details**	Sociodemographic characteristics, details of the care team
**Medical history**	Conditions and past medical procedures
**Medication**	Regular medication and medication to be taken as required
**Vaccinations**	COVID-19 and influenza vaccinations
**Allergies**	List of allergies
**Vital parameters and laboratory tests**	List of parameters with history
**Devices**	List of used medical devices (e.g., walker, pacemaker)
**Mental health**	List of mental health test scores, stress level and sleep problems with history
**Healthcare appointments**	List of appointments with reason and summary
**Health behaviour**	Physical activity, diet, smoking, substance use, functional limitations, activities of daily living, treatment burden and health plans (e.g., advance directives, emergency plan)
**Supervision by specialist team**	List of recommendations and summary
**Suicide Protocol**	For use in emergency situations and when indicated by mental health test scores

**NOTES**: * should be updated at every contact.

**Table 3 behavsci-15-00079-t003:** Description of the sample in the pilot study.

No.	Age	Sex (m/f)	No. of Chronic Conditions	Presence of Psychiatric Diagnosis (y/n)	HADS ^a^ Total Score at Baseline	HADS ^a^ Total Score at Follow-Up	No. of CM Contacts	Collaboration with PCP ^b^ (y/n)	Health Goals	Description of Goal Attainment
1	81	m	14	y	17	-	6	y (nephrologist)	(a) regulate fluid intake due to renal failure (b) improve sleep hygiene practises	(a) patient started to regulate fluid intake (b) not initiated because patient passed away unexpectedly during the intervention (cause of death unrelated to study)
2	71	f	12	n	17	21	20	y	(a) increase physical activity (b) start psychotherapy (c) engage in positive activities	(a) increased activity per day (b) started treatment (c) incorporated positive activities into daily routine by exploring self-care resources and old hobbies
3	69	f	10	y	14	10	4	y	(a) increase physical activity (b) confronting anxiety-inducing activities (c) building more resilience to external stressors	(a) increased daily walking distance (b) confronted anxiety-inducing activities through gradual exposure (c) regulated external stressors better at first, but experienced elevated distress due to newly diagnosed cancer
4	67	f	11	y	10	16	5	y	(a) start exercising (b) monitor sleep hygiene (c) increase social activities (d) improve depressive symptoms	(a) implemented daily short exercise sessions (b) perceived status quo as unchangeable at first, then took small steps towards improving sleep hygiene (c) discussed various ideas for increasing social activities, but none were implemented (d) started medication for improving depressive symptoms
5	72	f	10	n	29	-	2	y	(a) gain weight (b) improve stress management	(a) not initiated (b) distress was discussed but patient was not available after second contact
6	67	f	8	y	4	21	6	y	(a) improve digestive symptoms (b) improve muscle strength (c) improve conflict management skills (d) reduce oedema	(a) not achieved, maintained dietary log, nutritional counselling initiated (b) started occupational therapy (c) discussion of relaxation exercises (d) treatment of oedema monitored by clinical specialist team, minor improvement towards end of intervention
7	68	m	12	y	20	17	6	n	(a) eat a more nutritious diet (b) increase physical activity (c) lose weight	(a) measures were discussed and some of them implemented (b) started exercising regularly (c) not achieved
8	76	m	7	n	18	8	4	y	(a) increase physical activity (b) reduce stress related to caregiving responsibilities	(a) incorporated more exercises into daily routine (b) was advised to request an increase in the level of care, which was granted, and as a result received more support
9	65	m	12	n	16	13	7	y	(a) increase physical activity (b) lose weight	(a) started physical exercise (b) not achieved

**NOTES**: ^a^ HADS = Hospital Anxiety and Depression Scale ([60]), range: 0–42; ^b^ PCP = primary care provider.

## Data Availability

Research data are available from the corresponding author upon reasonable request.

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
