# Peer review of "Adapting and Implementing a Blended Collaborative Care Intervention for Older Adults with Multimorbidity: Quantitative and Qualitative Results from the ESCAPE Pilot Study"

_behavsci, 2025, doi:10.3390/bs15010079_

Round 1

Reviewer 1 Report

Comments and Suggestions for Authors

This study is a valuable contribution to the topic of person-centred care in multimorbidity. I have only a few edits and therefore have not provided an additional file. 

line 49 Add reference for this sentence. 

page 8 results. Please explain why the flowchart shows 6 included in the analysis, however table 3 includes data for the 9 included at baseline. Did you still analyse the baseline data for those excluded? if so, why?

line372 add 'ed' to implement. >implemented

Author Response

Comment: This study is a valuable contribution to the topic of person-centred care in multimorbidity. I have only a few edits and therefore have not provided an additional file. line 49 Add reference for this sentence. 

Response: Thank you for your positive feedback on our study. We added a few references for line 49 regarding the effects of physical-mental multimorbidity: http://dx.doi.org/10.5888/pcd17.200155; https://doi.org/10.1186/s12875-015-0344-4; https://doi.org/10.1186/s12875-023-02056-y 

Location of revisions: p. 2, Section 1, l. 52

Comment: page 8 results. Please explain why the flowchart shows 6 included in the analysis, however table 3 includes data for the 9 included at baseline. Did you still analyse the baseline data for those excluded? if so, why?

Response: Thank you for your valuable comment regarding this matter. We acknowledge the confusion surrounding the flowchart and Table 3. We have updated the flowchart to clearly show that our baseline and process analyses included data from 9 patients, while the qualitative analysis was conducted with data from 6 patients. The baseline data analysis was integrated with the analysis of intervention processes from the CMs’ documentation in the electronic registry, allowing us to include all patients in the study, even those who had unfortunately passed away or were unavailable for follow-up.

Location of revisions: p. 8, Figure 2

Comment: Line 372 add 'ed' to implement. >implemented

Response: We rephrased the sentence.

Location of revisions: Page 13, section 3.4, ll. 370-372

Reviewer 2 Report

Comments and Suggestions for Authors

Dear Authors,

the comments in the annex file.

Best

Author Response

Comment 1: Dear Authors, First of all, I would like to express my sincere gratitude for the opportunity to contribute my opinion to the evaluation of your manuscript. I found the topic addressed to be extremely interesting and highly relevant to our field. The research presents numerous useful and promising insights that could lead to significant advancements in our sector. I preliminarily conclude that the quality of the manuscript is already very high, and only minor adjustments could further enhance its impact: 

Response 1: Thank you for your careful consideration of our manuscript and for your positive evaluation of the study's design and report.

Comments 2: The title may be somewhat misleading, or at least that was my impression upon reading it. The phrase “Results from the ESCAPE Pilot Study” could benefit from the inclusion of terms such as “quantitative and qualitative” and “preliminary,” thereby giving readers a clearer indication of the dominant themes. For example: “Quantitative and Qualitative Results from the Preliminary ESCAPE Pilot Study.”

Response 2: We agree that the title can be more informative. We changed it to “Quantitative and Qualitative Results from the ESCAPE Pilot Study.”

Location of revisions: Title

Comment 3: In the introduction, the authors reference the Chronic Care Model (CCM); however, given the preventive aspect of the Case Manager’s role (at least in terms of complications), would it not be more precise to also refer to the Expanded Chronic Care Model (ECCM)? Additionally, further elaboration on the complexity of applying the same concepts to different healthcare models (lines 68–79) could enrich this section.

Response 3: 

We appreciate your feedback. We have expanded on the CCM and ECCM in the introduction and explained in more detail how BCC draws on these models. We also agree that the complexity of applying BCC in different health care models is very interesting and therefore an objective of the main ESCAPE trial, but could not be explored empirically in the pilot. To avoid raising false expectations in the reader, we have elaborated on this in the Discussion section instead of in the Introduction. We hope that the interested reader will turn to the main results paper of the ESCAPE trial (projected for 2026).

Location of revisions: Section 1, p. 2, 2nd and 3rd paragraph; Sections 4.2 and 4.3, pp. 14-15, ll. 444-497

Comment 4: Section 2.3 could be subdivided into subsections to facilitate reading. For example: 2.3.1 Procedure (lines 130-141) 2.3.2 Sample (lines 142-151) 2.3.3 Data (lines 152-157) 2.3.4 Quantitative Analysis

Response 4: Section 2.3 could be subdivided into subsections to facilitate reading. For example: 2.3.1 Procedure (lines 130-141) 2.3.2 Sample (lines 142-151) 2.3.3 Data (lines 152-157) 2.3.4 Quantitative Analysis

Location of revisions: Section 2.3, p. 6

Comment 5: Section 4.3 in the Discussion could be integrated and expanded upon earlier in the discussion, without being specifically titled “Comparison with the Literature,” as this aspect is implicitly addressed throughout the discussion. In expanding this section, I would suggest placing greater emphasis on the key figures the project is focusing on, such as the Nurse Case Manager, which is only briefly mentioned (and should be supported by an expanded bibliography). This could be framed within a preventive medicine perspective, such as Lifestyle Medicine. There are already interesting studies on this topic in the literature, and I hope the authors will consider the potential improvements that could be derived from professionals oriented toward such a culture in the development of their prestigious project. In summary, the manuscript presents results of notable scientific interest, but it could benefit from some improvements to make it even more appealing to the scientific community. Honestly, I will recommend a Major Revision (though it likely merits Minor Revision) with the hope that the authors will appreciate the comments and reflections provided. Best regards and good work.

Response 5: We appreciate your feedback regarding the need for more specificity in the section titled 'Comparison with the Literature.' In response, we have restructured sections 4.2 and 4.3, renaming section 4.3 to 'Practical Implications and Future Directions.' Additionally, we have expanded our discussion of the CM role, focusing on two significant challenges identified in both the literature and our study: the necessity for comprehensive training for CMs—covering psychosocial support, strategies for addressing behavioral health aspects, and a thorough understanding of the healthcare system—and the integration of the CM into patients’ healthcare networks. We also added some relevant references.

Location of revisions: Sections 4.2 and 4.3, pp. 14-15, ll. 444-497; Section 5, ll. 1st sentence

Reviewer 3 Report

Comments and Suggestions for Authors

The concept of having CM to communicate plans with patient and PCP is commendable.

The LOGIC model is nicely presented.

This pilot study is affected by low number of participation. Only 9 completed baseline data and analysed. Do you think it is feasible to pilot this in five EU states when the single site pilot received such a low number. What did not went well? I believe this is the major aspect to be addressed before the ESCAPE trial could start.

Qualitative input from n=6 could hardly yield holistic views from users.

Design: 3-month goal attainment is quite achievable but may not be sustained thereafter. Maybe can consider follow-up for 12 months so to know if this is feasible on the patients and create meaningful change in lifestyle.

Table 2. The last column was too generic. Lacking quantitative descriptions on the physical activities (time, duration). Increased and decreased are crude descriptors. And were these self-reported? is there any objective measurements such as wearable devices?

Why the HADS score is not measured again at the end of 3months and compared against baseline?

3.2 Delivery. 12 bi-weekly multidisciplinary discussions can be labor intensive depending on the size of the team. And it doesn't quite spread out, as in the discussions will end at 6 weeks? whereas this is a 3-month follow-up program?

Author Response

Comment 1: The concept of having CM to communicate plans with patient and PCP is commendable. The LOGIC model is nicely presented. This pilot study is affected by low number of participation. Only 9 completed baseline data and analysed. Do you think it is feasible to pilot this in five EU states when the single site pilot received such a low number. What did not went well? I believe this is the major aspect to be addressed before the ESCAPE trial could start. Qualitative input from n=6 could hardly yield holistic views from users.

Response 1: Thank you for your feedback, which we hope to be able to address to your satisfaction. The main ESCAPE trial aims to recruit the majority of participants (N=124 out of a total of N=300) from Germany. We have therefore decided to run a small pilot study in Germany only to test the feasibility of our recruitment strategy and to inform any necessary adjustments to the intervention. Due to funding structures and limited resources, the scope of the pilot test was kept deliberately small, with a target recruitment of 10 patients. Overall, recruitment was successful; of the 44 patients approached, 10 agreed to participate and 90% of these met the eligibility criteria for inclusion in the pilot. This information has been added to the Methods section. In our limitations, we have highlighted that these results are preliminary and need to be confirmed in our ongoing ESCAPE trial. We also plan to conduct a comprehensive 360-degree process evaluation of the main trial, integrating electronic documentation, patient-reported experience measures, and qualitative interview data from all members of the BCC team, including CMs and GPs. These future plans have now been included in the Discussion section.

Location of revisions: Section 2.3.2, p. 6, ll. 167-170; Discussion, 4.3, p. 15, ll. 500-507

Comment 2: Design: 3-month goal attainment is quite achievable but may not be sustained thereafter. Maybe can consider follow-up for 12 months so to know if this is feasible on the patients and create meaningful change in lifestyle.

Response 2: We agree, but the primary aim of the pilot was to assess whether patients would accept this format and be able to tackle any goals. In the main trial, the intervention will last nine months, with follow-up assessments at the end of the intervention and again six months and longer thereafter to check for sustained changes. This information has been incorporated into the Discussion section

Location of revisions: Discussion, section 4.1, pp. 13-14, ll.415-432; section 4.3, p. 15, ll. 500-507

Comment 3: Table 2. The last column was too generic. Lacking quantitative descriptions on the physical activities (time, duration). Increased and decreased are crude descriptors. And were these self-reported? is there any objective measurements such as wearable devices?

Response 3: The description of health goals and goal attainment was based on the CMs' documentation in the electronic registry. We have specified this under Methods. We do not plan on using measurements from wearables. We agree that the use wearables would add an objective measure. However, based on our previous experience, data safety requirements in several European countries are different and in part very strict, so that it is not feasible for our trial design plus exceeds the tight budget. Also, in the pilot study, these measurements could not be indicative of significant changes due to the limited sample size.

Location of revisions: Section 3.2, p. 11, ll. 279-280

Comment 4: Why the HADS score is not measured again at the end of 3months and compared against baseline?

Response 4: We added the HADS scores at follow-up to the table and Results section. 

Location of revisions: Table 3, pp. 9-10; section 3.2, p. 11, ll. 288-291

Comment 5: 3.2 Delivery. 12 bi-weekly multidisciplinary discussions can be labor intensive depending on the size of the team. And it doesn't quite spread out, as in the discussions will end at 6 weeks? whereas this is a 3-month follow-up program?

Response 5: We apologise for any confusion. Our initial plan was to conduct bi-weekly case review meetings; however, due to the limited sample size, this frequency proved unnecessary. We agree that these meetings are labor-intensive and require significant staff resources, which led to the cancellation of some scheduled meetings. Bi-weekly meetings are only necessary with a sufficiently large patient load in the intervention. We have corrected this detail in the Results section.

Location of revisions: Section 3.2, p. 11, l. 279

Reviewer 4 Report

Comments and Suggestions for Authors

Hello,

I appreciate the opportunity to review the article on such an important topic.

The article is well written, in a clear and concise manner, allowing for easy reading and understanding of the topic.

The details of the trial and its results are presented clearly, allowing for learning and application by the reader.

One small suggestion at the beginning of the discussion: it is recommended to consider adding a paragraph stating the objectives. 

Well done!

Author Response

Comment: 

Hello,

I appreciate the opportunity to review the article on such an important topic.

The article is well written, in a clear and concise manner, allowing for easy reading and understanding of the topic.

The details of the trial and its results are presented clearly, allowing for learning and application by the reader.

One small suggestion at the beginning of the discussion: it is recommended to consider adding a paragraph stating the objectives. 

Well done!

Response: We appreciate your positive feedback and the time you dedicated to reviewing this manuscript. Following your helpful suggestion, we have included a statement of the study's objectives at the beginning of the discussion.

Location of revisions: Section 4, page 13, ll. 407-413

Round 2

Reviewer 2 Report

Comments and Suggestions for Authors

Dear Authors,

the Editor will be conducted in the publication.

Best